# Primary Mucosa-Associated Lymphoid Tissue Lymphoma of the Parotid Gland in 32-Year-Old Male, a Case Report

**Bianca M. Glass** [1] ⬤, **Mira Al Jaberi** [2], **John H. Irlam** [3] **and Samir M. Dalia** [4,*]

1   College of Osteopathic Medicine, Joplin Campus, Kansas City University, 2901 St. Johns Blvd, Joplin, MO 64804, USA
2   College of Medicine, University of Jordan, Amman 11942, Jordan
3   Ferguson Medical Laboratories, 1801 W 32nd St, Joplin, MO 64804, USA
4   Department of Medical Oncology, Mercy Hospital, 100 Mercy Way, Joplin, MO 64804, USA
*   Correspondence: samir.dalia@mercy.net

**Abstract:** Primary lymphomas of the salivary gland are rare. The most common subtype is MALT lymphoma. MALT lymphoma has an indolent clinical course, and patients often present with a prolonged history. Evaluations of parotid masses begin initially with radiological imaging, but pathological and histological examination remains the mainstay of definitive diagnosis. This case describes a primary non-Hodgkin's lymphoma of the parotid gland in a healthy 32-year-old male. This case report will evaluate the prevalence of primary MALT lymphoma and discuss the possible presentation.

**Keywords:** MALT; NHL; non-Hodgkin Lymphoma; parotid gland tumor; salivary lymphoma; sialolithiasis; Sjogrens; CD20; B-cell lymphoma; IgD

## 1. Introduction

Non-Hodgkin lymphoma (NHL) accounts for the seventh most prevalent cancer that has a high mortality rate and has increased in incidence exponentially in the past 4 decades. Within the NHL class type exists over 40 different subtypes, including B-cell, T-cell, and NK proliferation that lack Reed–Sternberg cells and CD15/CD30 in the histology. Risk factors for NHL include occupational exposures, associations with autoimmune diseases, male sex, infectious causes, prior radiation, and lifestyle factors. While primary lymphomas of the salivary gland are rare, the most common subtype is marginal-zone B-cell mucosa-associated lymphoid tissue-type lymphoma, MALT. Based on the World Health Organization (WHO4) Classification of hematolymphoid tumors, mucosa-associated lymphoid tissue (MALT) lymphoma is defined as "an extranodal lymphoma composed of morphologically heterogenous small B-cells, including marginal zone cells, cells resembling monocytoid cells, small lymphocytes, and scattered immunoblasts and centroblast-like cells" [1]. This definition has not changed in the latest WHO update (WHO5) and is also similar to the latest International Consensus Classification (ICC) [2]. MALT lymphomas can arise in the gastric mucosa, lungs, eyes, skin, thyroid, and salivary glands [3]. MALT lymphoma has an indolent clinical course before disseminating, and patients often present with a prolonged history [4,5]. The treatment of non-gastric MALT lymphomas is dependent on its staging and the Ann Arbor system [5]. This case report describes a 32-year-old male presenting a 2-year history of presumed sialolithiasis with pathology revealing primary MALT lymphoma of the parotid gland. This case represents the importance of utilizing imaging modalities and surgical intervention for the early diagnosis of MALT lymphoma.

## 2. Case Presentation

A 32-year-old, Caucasian male presents with a two-year history of a right-side cheek mass and no other symptoms. The patient indicates no past medical history, no past surgical history, no family history, and takes no medications. In 2021, he presented to his

primary care with a complaint of a right-side parotid mass and no other symptoms and was diagnosed with sialolithiasis; however, upon recurrence and unremarkable labs, the primary care physician ordered a CT scan. In 2021, a CT indicated a diffusely enlarged, heterogeneous parotid mass with diffuse, punctate, and coarse calcifications. Additionally, there was asymmetric enlargement of the right submandibular gland but no evidence of calcification in the duct. Further, some small cervical chain lymph nodes were noted. While the patient denied SICCA symptoms at the time, the CT scan corroborated potential Sjogren's syndrome. No further workup was performed, and the patient was lost to follow-up.

The patient presented at the ENT clinic for another evaluation with an otolaryngologist two years later with a right-side mass, which he claimed had increased in size. Given concerns that this could be a rare presentation of sarcoidosis, Ace levels were measured and found to be normal. The patient denied positivity for HIV or HBV from past workups. Essentially, no other workup other than surgical excision was performed. An excisional biopsy was performed taking a 2 × 1.2 × 2 cm piece from the right parotid gland. The pathology listed below demonstrated how the small sample size biopsied from the patient's parotid gland had no normal parotid tissue visible, as seen in Figure 1.

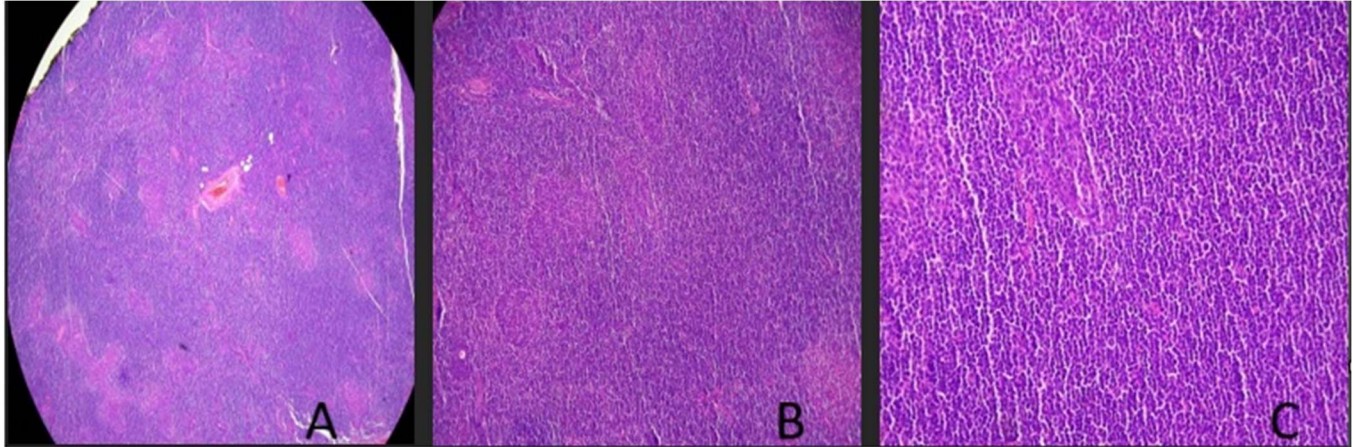

**Figure 1.** Parotid gland slices shown at various magnifications. (**A**) Low-power magnification demonstrates a diffuse lymphoid infiltrate with effacement of nodal architecture including lack of follicles and germinal center formation. From this parotid gland sample, clear demarcation of lymphoma from salivary gland tissue is not distinguished in the biopsy specimen. (**B**) Mid-power examination demonstrates a dense lymphoid infiltrate with a proliferation of monotonous-appearing lymphocytes. (**C**) High-power examination demonstrates a proliferation of small lymphocytes with overall mature-appearing chromatin and scant cytoplasm. No large atypical cells or areas of necrosis are identified.

Histopathology of two HE-stained slides demonstrated fragments of relatively dense lymphoid stroma with architectural distortion. Follicles and well-formed germinal centers were not readily apparent, as seen in Figure 1. There were questionable glandular elements present. The immunohistochemical staining initial pathology indicated that there were CD3/CD5-positive T-cells with increased numbers of CD-20/PAX 5-positive B-cells, with the occasional cell positive for CD30. Additionally, there was focal BCL2/BCL6 co-expression. CD23 highlights follicular dendritic cells. There was no overexpression of CD10 or Cyclin D1. The KI-67 proliferation index was low. The pan-cytokeratin stain highlighted that occasional glandular epithelial elements were present. Given the initial pathology report and the possibility of a lymphoproliferative process, a recommendation for additional hematopathology was performed.

The addended hematopathology report indicated, through a B-cell clonality panel, PCR rearrangement of the immunoglobulin heavy chain IGH and immunoglobulin Kappa chain IGK genes. The overall morphologic findings and lack of CD-5 or CD-10 expression

by atypical B-cells favored the diagnosis of MALT lymphoma in this asymptomatic 32-year-old with no prior medical history. Given these findings, the patient had an ANA workup with ANA positivity of 1:1280 with a nuclear-speckled pattern, anti-smith positivity, anti-Ro/SSB and anti-La/SSB negativity, dsDNA negativity, and SCL-70 negativity. His LDH was in a normal range, and his Beta 2 microglobulin was high normal. Hepatitis and HIV panels were all negative. The patient will be treated with radiation in an academic setting.

## 3. Discussion

This case report describes a rare presentation of a primary MALT lymphoma in the setting of a healthy young male with no history of autoimmune disease. Our patient is a 32-year-old male presenting with a 2-year history of presumed sialolithiasis with pathology revealing primary MALT lymphoma of the parotid gland. This case represents the importance of utilizing imaging modalities and surgical intervention for the early diagnosis of MALT lymphoma. Malignant lymphomas (MLs) represent a group of various diseases that arise from the clonal proliferation of lymphocytes. MLs are divided into Hodgkin lymphoma (histologically characterized by the presence of Reed–Sternberg cells and CD15 and CD30 staining) and non-Hodgkin lymphoma [3]. According to the latest World Health Organization (WHO) classification, the most common NHL is DLBCL [3]. Lymphoma of the parotid gland is a relatively rare occurrence among head and neck tumors. Around 1–4% of patients undergoing parotidectomy are found to have lymphomas, with the majority classified as NHL extra-nodal marginal-zone B-cell lymphomas, this can be demonstrated in Table 1, above. Mucosa-associated lymphoid tissue (MALT lymphoma) is an NHL and can occur in a variety of organs, including the orbit, conjunctiva, salivary glands, skin, thyroid glands, lungs, stomach, and intestine [4]. A retrospective study including 31 patients diagnosed with lymphoma of the parotid gland showed that 18 patients had MALT lymphoma (60% of patients), making it the most common subtype of primary parotid lymphoma [5]. MALT lymphomas usually arise in a setting of chronic inflammation caused by infectious agents or an autoimmune disorder such as Sjogren's syndrome, which was ruled out in our patient [2]. It has been recognized that constant antigenic stimulation from prolonged inflammation causes B-cell-receptor signaling and leads to the activation of NF-kB to induce the proliferation of B-cells, which eventually undergo malignant transformation [6].

**Table 1.** Salivary gland tumors [3,7–11].

| Tumor Type | Incidence Worldwide (per 100,000) | Percentage Occurring in the Parotid Gland (%) | Prognosis |
|---|---|---|---|
| Benign salivary gland tumors | 0.4–13.5 | ~80 | Generally excellent, most are curable with surgery |
| Malignant salivary gland tumors | 0.5–3 | ~64–80 | Varies based on tumor characteristics; early detection is key for better prognosis |
| Lymphomas of salivary glands (MALT, follicular, DLBCL) | Very rare 1.7–3.1% of all salivary gland neoplasms | 79 | Dependent on the stage |
| MALT lymphoma of salivary gland | Very rare | 5–10 | Generally favorable, typically low-grade, slow-growing |

Ethnic variation and racial disparities have been found in non-Hodgkin lymphoma incidence [12]. The risk of NHL in Caucasian populations remains unclear and is an inter-

esting area for further research. Evaluations of parotid masses should include magnetic resonance imaging (MRI) or CT to determine tumor size, shape, and location [4]. Definitive diagnosis is made through pathological and histological examination. No specific immunohistochemical marker has been identified for MALT lymphoma, but the utilization of a panel of immunochemical stains is essential for diagnostics. The presence of CD-20 positivity, a B-cell marker between glands, and reactive follicles will usually indicate a specimen consistent with lymphoma. The utilization of CD21 and CD10 will further characterize lymphoid follicles and IgD positivity or negativity for mantel versus marginal zone distinction. The negativity of lymphoma cells for IgD, CD5/10, BCl6, and cyclin D1 helps to exclude small-B-cell lymphomas. The defining feature of these markers can be found in Figure 2.

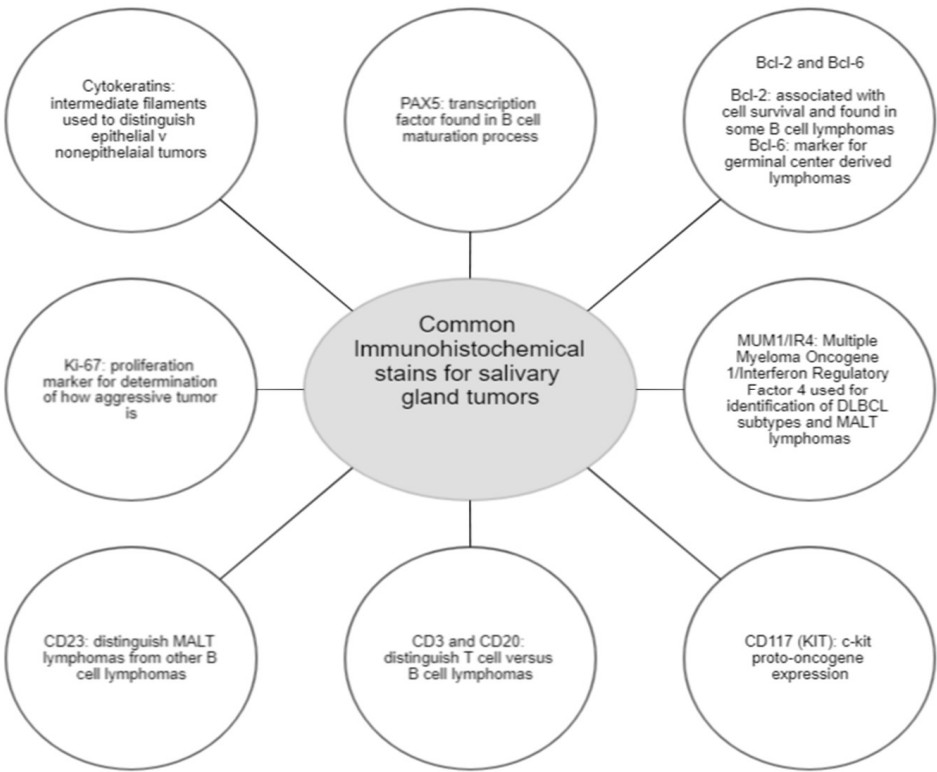

**Figure 2.** Immunohistochemistry of salivary gland tumors [3,7–9,13].

MALT lymphomas of the parotid gland are usually deemed to be indolent and, therefore, borderline lesions; however, when characterized by destructive cavitation lesions with reaction follicle replacements, MALT lymphomas are said to be more established and, therefore, concerning. Analyses of tissue samples show that B-cell monoclonality is highly correlated with the presence of the morphological features of early MALT [14]. The utilization of IGH and IGK gene arrangements through B-cell panels via PCR is highly correlated with the detection of marginal cells, MALT, DLBCL, and CLL. The utilization of this rearrangement, in addition to other marker positivity, helps deduce the presence of MALT versus another type of NHL [8].

The staging standard used by most hematologists in extra-nodal lymphoma cases is the Ann Arbor staging system [5]. According to the National Comprehensive Cancer Network guidelines, the treatment of MALT lymphoma varies and is based on the Ann Arbor staging system [15]. After surgical resection and immunochemical diagnostics, staging using PET-CT will determine if treatment such as rituximab or the adjunct of radiation and chemotherapy is necessary. Conventional chemotherapeutic agents such as cyclophosphamide, vincristine, and prednisone have been used in combination with rituximab. Treatment does not always have to be given because marginal lymphoma

usually has an indolent clinical course [13]. The monitoring of LDH levels can be utilized in the determination of therapeutic effects and favorable prognoses. A recurrence rate for parotid MALT tumors can occur, and salvage surgery, radiation, or chemo-immunotherapy can be used [8].

Primary malignant lymphomas of the parotid gland are very rare, especially in young patients with no past medical history, as seen in Table 1. We describe a case of primary non-Hodgkin lymphoma of the parotid gland arising in a healthy, young, 32-year-old male. To the best of our knowledge, there is only one case similar to our patient in the literature [12,16]. A previously healthy 54-year-old woman presented with a progressively and slowly growing right parotid mass; further investigations revealed the diagnosis of MALT lymphoma of the parotid gland in the absence of Sjogren's syndrome [12]. In comparison with our case, both patients had no history of Sjogren's syndrome. However, in contrast to our case, MALT lymphomas are twice as likely to occur in females and are more common in patients above 50, making our case unusual [12]. The 32-year-old male presented in this case report had no ANA or antibody testing nor symptoms consistent with Sjogren's. Additionally, he had no history of autoimmune disease or a family history of autoimmune diseases. It is possible he had Sjogren's, which could have predisposed him to MALT lymphoma, as the risk is 44-fold higher in patients with a background of Sjogren's [17]. However, with a negative Anti-SSA/Ro and Anti-SSB/La antibodies, this situation seems to be more consistent with a primary tumor of an idiopathic nature in this 32-year-old male.

While marginal-zone lymphomas can be found in oncology clinics, they are generally found in lymph nodes and are not extra-nodal, accounting for a small percentage of non-Hodgkin lymphomas. The most commonly documented extra-nodal site is the stomach, with an incredibly low incidence rate. Additionally, the average median age appears to be the sixth decade of life, with a predilection toward females, making this presentation of a male in his third decade more intriguing [9]. Should a physician suspect MALT, an EGD should be performed to rule out gastric involvement, and Anti-SSA/Ro and Anti-SSB/La antibodies should be tested to rule out Sjogren's syndrome as the propagator of its existence. Additionally, if primary symptoms begin in the ocular adnexa, orbital and salivary MRI or CT should be performed if clinically indicated. Chlamydia psittaci can be tested in the tumor biopsy sample as an antigenic cause of ocular MALT with simultaneous salivary gland involvement. Staging risks and workups should include LDH, beta2 microglobulin, HCV-RNA, HBV, and HIV, in addition to full-flow cell cytometry. While it is indolent in nature, recommendations of follow-up EGDs for extra-nodal MALT lymphomas and recommendations of PETs are controversial [11].

**Author Contributions:** Conceptualization, B.M.G.; methodology, B.M.G. and S.M.D.; validation, B.M.G., S.M.D. and M.A.J.; formal analysis, B.M.G. and J.H.I.; investigation., S.M.D. and M.A.J.; resources, B.M.G. and J.H.I..; data curation, B.M.G., S.M.D. and M.A.J.; writing—original draft preparation, B.M.G. and M.A.J.; writing—review and editing, B.M.G., S.M.D., J.H.I and M.A.J.; visualization, B.M.G.; supervision, B.M.G. and S.M.D.; project administration, B.M.G. and S.M.D.; All authors have read and agreed to the published version of the manuscript.

**Funding:** This research received no external funding.

**Institutional Review Board Statement:** Not applicable.

**Informed Consent Statement:** According to §46.104, Exempt research by use of Secondary research for which consent is not required: However, Informed consent was obtained from the patient for the above diagnostic evaluation, treatment, and cross-communication for tertiary care.

**Data Availability Statement:** Data sharing not applicable.

**Conflicts of Interest:** The authors declare no conflict of interest.

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
