# Peer review of "Primary Mucosa-Associated Lymphoid Tissue Lymphoma of the Parotid Gland in 32-Year-Old Male, a Case Report"

_2673-8937, doi:10.3390/ijtm3040030_

Round 1

Reviewer 1 Report

An atypical case of salivary gland lymphoma in a young male without clinical symptoms and family history is described. The clinical case is interesting. However, I think that the authors should add CT images for illustration, which may contribute to a more successful diagnosis if such a patient is encountered in clinical practice in the future. Of course, in the case of the salivary glands, the oncological awareness of the doctor in relation to MALT lymphoma is low, but it is useful to formulate clinical recommendations for differential diagnosis at the end of the Discussion section.

Author Response

We have looked at the reviewers comments and have made appropriate changes to our manuscript.   We wanted to highlight the following. 

We did include a differential diagnosis section to help clinicians better improve their understanding on how to approach a workup for these tumors.   While we tried to get imaging and histological slides we were unable to get these from the case due to technical issues.   We do not feel they add much more to the case report.    All of the initial workup was added to the case.   It was very limited as the patient was seen by ENT and then decided to pursue follow up at a tertiary care center.

Reviewer 2 Report

This is a case report of a primary parotid gland MALT lymphoma of a 32 year old male. The case report is well written, and it is easy to read, and to understand. Although it may be unusual neoplasia in that location, MALT lymphomas are seen frequently in hematopathology department worldwide, and parotid gland are also seen. So it is no exceptional case. But this should not prevent the authors to publish.

The text could be improved with:

1) A better description of MALT lymphoma from a hematopathologist point of view. Please use as reference the WHO 4th edition, the International Consensus Classification of Mature Lymphoid Neoplasms 2022 (https://doi.org/10.1182/blood.2022015851), and/or beta version of WHO5.

2) Could you please show images, histological, CT, etc.

3) Clinical variables of some of the markers, hematological, serologica, could be detailed.

4) The authors may improve the differential diagnosis with the other subtypes of lymphoma making a table.

5) Please explain why this is an exceptional case.

Author Response

(The authors gave the same response as above.)
